# The Effect of the *ortho* Nitro Group in the Solvolysis of Benzyl and Benzoyl Halides

**DOI:** 10.3390/ijms20164026

**Published:** 2019-08-18

**Authors:** Kyoung-Ho Park, Chan Joo Rhu, Jin Burm Kyong, Dennis N. Kevill

**Affiliations:** 1Department of Chemical and Molecular Engineering, Hanyang University,Ansan-si, Gyeonggi-do 15588, Korea; 2Korea Conformity Laboratories, Seoul 08503, Korea; 3Department of Chemistry and Biochemistry, Northern Illinois University, DeKalb, IL 60115, USA

**Keywords:** *o*-nitrobenzyl bromide, *o*-nitrobenzoyl chloride, *ortho* nitro group, Grunwald–Winstein equation, intramolecular nucleophilic assistance, solvolysis, nucleophilicity, ionizing power

## Abstract

A kinetic study was carried out on the solvolysis of *o*-nitrobenzyl bromide (*o*-isomer, **1**) and *p*-nitrobenzyl bromide (*p*-isomer, **3**), and *o*-nitrobenzoyl chloride (*o*-isomer, **2**) in a wide range of solvents under various temperatures. In all of the solvents without aqueous fluoroalcohol, the reactions of **1** were solvolyzed at a similar rate to those observed for **3**, and the reaction rates of **2** were about ten times slower than those of the previously studied *p*-nitrobenzoyl chloride (*p*-isomer, **4**). For solvolysis in aqueous fluoroalcohol, the reactivity of **2** was kinetically more reactive than **4**. The *l/m* values of the extended Grunwald–Winstein (G–W) equation for solvolysis of **1** and **2** in solvents without fluoroalcohol content are all significantly larger than unity while those in all the fluoroalcohol solvents are less than unity. The role of the *ortho*-nitro group as an intramolecular nucleophilic assistant (internal nucleophile) in the solvolytic reaction of **1** and **2** was discussed. The results are also compared with those reported earlier for *o*-carbomethoxybenzyl bromide (**5**) and *o*-nitrobenzyl *p*-toluenesulfonate (**7**). From the product studies and the activation parameters for solvolyses of **1** and **2** in several organic hydroxylic solvents, mechanistic conclusions are drawn.

## 1. Introduction

Generally, *ortho*-substituted benzyl halides solvolyze somewhat more slowly than their *para-*isomers since nucleophilic displacement at the reaction center is reduced by the steric hindrance of a substituent group in the *ortho*-position [1,2]. However, the *o*-isomer (*o*-carbomethoxybenzyl bromide, **5**) in various solvents is known to undergo solvolytic reactions many times more rapidly than its the *p*-isomer, e.g., in the case of carbomethoxybenzyl bromide (**6**) [3]. Relatively higher reactivity of the *o*-isomer (**5**) has been explained on the assumption that its carbomethoxylic group can participate electronically in the activation process by releasing electrons to the vacant *p*-orbital developing at the reaction center. We have previously reported our studies using the Grunwald–Winstein equation in investigations of the reaction mechanisms for the solvolyses of **5** and **6**, which were known to have the possibility of intramolecular nucleophilic attack at the α-carbon (Equation (1)) [3].

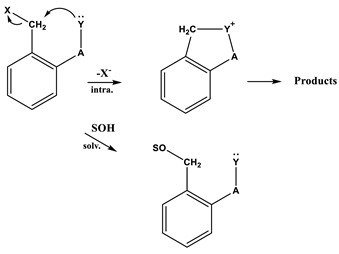
(1)

In the consideration of the solvolyses of benzoyl chloride and *p*-nitrobenzoyl chloride (**4**) [4], they have been extensively studied and the available kinetic data were fairly analyzed using both the simple (one-term) Grunwald–Winstein Equation (2) and the extended (two-term) Grunwald–Winstein Equation (3) [5,6,7,8,9,10,11].
log (*k* / *k*_o_) = *mY*_X_ + *c*,(2)
log (*k* / *k*_o_) = *lN*_T_ + *mY*_X_ + *c*.(3)

In Equations (2) and (3), *k* and *k*_o_ represent the rate constants of solvolysis in a given solvent and in 80% ethanol (EtOH), respectively; *l* is the sensitivity to changes in solvent nucleophilicity (*N*_T_) [7,8]; and *m* is the sensitivity to changes in solvent ionizing power (*Y*_X_) [9,10,11]. Application of Equations (2) and (3) showed a balance between addition–elimination (association–dissociation, Equation (4)) and ionization (Equation (5)) mechanisms. For solvolyses of **4** [4], all of the solvents except the highly ionizing and low nucleophilicity solvents (i.e., 97% 1,1,1,3,3,3-hexafluoro-2-propanol, HFIP) gave a good correlation with a high *l* value and a moderate *m* value (*l**/**m* =3.3), consistent with the operation of the addition–elimination mechanism (Equation (4)). The solvolysis of benzoyl chloride has led to the detection of both ionization (*l**/**m* = 0.59) and addition–elimination pathways (*l**/**m* = 2.8). Bentley and co-workers also carried out a series of studies on the solvolyses of benzoyl chloride and derivatives. For the parent compound and several of the derivatives, a mechanism involving an *S*_N_2 character with bond breaking running ahead was proposed for highly aqueous solvents and a carbonyl addition mechanism in less aqueous solvents [12,13,14,15].

Addition-elimination mechanism for benzoyl chloride in less ionizing and more nucleophilic solvents:
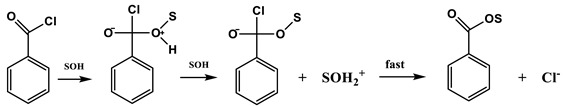
(4)

Ionization mechanism for benzoyl chloride in more ionizing and the least nucleophilic solvents:
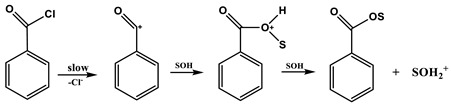
(5)

We previously reported on the solvolyses of **5** and **6** in the wide range of solvents [3], and partially on the solvolysis of *o*-nitrobenzyl bromide (**1**) in 80% EtOH, 50% EtOH, and 97% and 70% trifluoroethanol (TFE) [16]. The solvolyses of **5** and **6** were found because of the internal assistance rather than the steric hindrance of the *ortho* position on the attack on the alkyl carbon from the presence of the *ortho* carbomethoxy group, to undergo solvolysis by a duality of mechanisms, i.e., intermolecular assistance from a solvent molecule and intramolecular assistance). Andrews and co-workers studied the effects of *ortho*-substitution on nucleophilic substitution reactions of benzylic and benzhydrylic derivatives [1,2]. They proposed that *ortho*-groups possessing a nucleophilic center could give intramolecular assistance to reactions. In spite of intensive experimental examination of the mechanisms of *ortho*-substituted benzyl halides, the mechanisms of these reactions have been less studied in kinetics than their *para*-isomers. Accordingly, a study of the reaction mechanism for **1** under solvolytic conditions is of interest as a simple model. In particular, the kinetics and mechanism of solvolysis of *o*-nitrobenzoyl chloride (**2**) have not been studied.

In a better attempt to define the reaction pathways for reactions of **1** and **2** under solvolytic conditions, we have applied the Grunwald–Winstein Equation (Equations (2) and (3)) to the solvolyses of **1** and **2** in a variety of pure and binary solvents. Mechanistic conclusions were then drawn from a consideration of the analyses using the extended Grunwald–Winstein equation, including a comparison with the *l* and *m*-values previously determined from kinetic studies of other substituted benzyl and benzoyl halides. We have investigated the products from solvolysis of **1** and **2** in various solvents by HPLC and GC/mass analysis (all corresponding results are in the section of Appendix A), together with calculated enthalpies and entropies of activation. The solvolyses of *p*-nitrobenzyl bromide (**3**) and **4** were also studied in order to compare the isomeric difference between *o*- and *p*-isomers.

## 2. Results and Discussion

The rate constants of solvolysis of **1** and **3** at 45.0 °C, and **2** at 25.0 °C, were measured in a variety of solvents and presented in Table 1, together with the *k*_(**1**)_/*k*_(**3**)_ and *k*_(**2**)_/*k*_(**4**)_ ratios. Literature values for solvolytic rate constants of **4** at 25.0 °C were used to calculate *k*_(**2**)_/*k*_(**4**)_ ratios [4,12,13,14,15]. The enthalpies and entropies of activation were calculated from the rate constants of solvolysis of **1**, **2,** and **3** reported in Table 2 and Table 3.

The *ortho-*substituted compounds (benzyl halides) [1,2] are generally less reactive than their *p*-isomers, although the reactivity differences are not large. The comparatively low reactivities of the *o*-isomers may result because of steric hindrance by the ring substituents to solvation of the exocyclic carbon atom in the transition state. However, in Table 1, the rate constant ratios (*k*_(**1**)_/*k*_(**3**)_) for solvolysis of the *o*-isomer (**1**) and *p*-isomer (**3**) are almost identical in all of the solvents except aqueous TFE at 45.0 °C, and somewhat more rapidly in aqueous TFE. The *k*_(**1**)_/*k*_(**3**)_ ratios are similar to those previously reported within the analyses of solvolysis of **5** and **6** in all of the solvents [3], i.e., the *o*-nitro group provides internal assistance in the displacement reaction of benzyl bromide like the *o*-carbomethoxylic group. Andrews and coworkers [1,2,3] have demonstrated that the greater reactivity of the *o*-isomer is attributed to internal participation (intramolecular nucleophilic assistance) by a neighboring carbomethoxylic group (Equation (1)). Such participation is not possible for the *p*-isomer because of unfavorable molecular geometry.

In Table 1, the reactivity of *o*-nitrobenzoyl chloride (**2**), unlike that of *o*-nitrobenzyl bromide (**1**), is appreciably less reactive than that of its *p*-isomer (**4**) in all of the solvents except aqueous fluoroalcohol, while that of **2** in aqueous fluoroalcohol is much greater (*k*_(**2)**_/*k*_(**4**)_ ≈ 14~2.2×10^4^) than that of its *p*-isomer. In other words, the lesser reactivity of the solvolysis of **2** in high nucleophilicity and low ionizing solvents, as introduced above, is consistent with that of the nucleophilic solvent assistance at a reaction center (acyl carbon) being reduced by the steric hindrance of the *o*-nitro group. However, the greater reactivity for **2** in the relatively low nucleophilicity and high ionizing solvents provides good evidence for other mechanistic changes.

The product studies to investigate the role of the *ortho* substituent for the solvolyses of **1** and **2** were followed by HPLC and/or LC/mass, respectively. The product ratios from solvolysis of **1** in 50% EtOH after 10 half-lives (ca. 26 h) at 45.0 °C were determined to be *o*-nitrobenzyl alcohol:*o*-nitrobenzyl ethyl ether:*o*-nitrobenzyl bromide:*o*-nitrosobenzaldehyde = 45.8%:41.1%:10.9%:0.635%, as previously published [3]. However, the product for **1** in 100% EtOH was not detected to be *o*-nitrosobenzaldehyde. The solvolysis products in previous research on **5** and *o*-nitrobenzyl *p*-toluenesulfonate (**7**) were also found to be phthalide and *o*-nitrosobenzaldehyde, respectively, showing intramolecular participation [3,17,18,19].

The product ratios from solvolysis of **2** in 100% EtOH after 21 min at 25.0 °C were determined to be ethyl-*o*-nitrobenzoate:*o*-nitrobenzoyl chloride = 97.4%:2.6%, and after 31 and 150 min only ethyl-*o*-nitrobenzoate was observed. For reaction in 70% TFE after 21 min, *o*-nitrobenzoic acid (45.3%) and trifluoroethyl-*o*-nitrobenzoate (54.7%) were obtained. In particular, the product studies for **2** were no evidence for an *o*-nitrosobenzene ion like *o*-nitrosobenzaldehyde which would have been formed by the reaction of intramolecular nucleophilic substitution in hydroxylic solvents of **1 [17,18,19]**. Therefore, evidence is presented that an *o*-nitro group participates as an internal nucleophile (intramolecular assistance) in promoting the solvolysis of **1** in binary organic solvents while that for the solvolysis of **2** (i.e., intramolecular assistance between the *o*-nitro group and the acyl carbon) does not.

A useful test in considering the detailed mechanisms of solvolysis of **1**, **2**, and **3** is to carry out a correlation analysis using the extended G-W Equation (3) and to compare the *l* and *m* values (*l/m* values) with those previously for **5** [3] and **7 [16]**. A consideration in terms of the extended G–W Equation (3) to the rate constants of solvolysis of **3** (from Table 1) leads to a good linear correlation with values of 1.19 ± 0.03 for *l*, 0.55 ± 0.02 for *m*, 0.09 ± 0.02 for *c*, and 0.995 for the correlation coefficient (*R*) (Figure 1). These values (*l* and *m*) are similar to those obtained to reflect the bimolecular mechanism within the analyses of the solvolyses of **6** (*l* = 1.17 ± 0.05, and *m* = 0.57 ± 0.03 in 19 solvents) **[3]** and *p*-nitrobenzyl *p*-toluenesulfonate (**8**, *l* = 1.04 ± 0.03, and *m* = 0.65 ± 0.02 in 34 solvents) [20,21]. In contrast to the good overall correlation observed for solvolyses of *p*-isomer (**3**), application of the extended G-W Equation (3) to 21 solvolyses of *o*-isomer (**1**) leads to a rather unsatisfactory correlation, with values of 0.66 ± 0.07 for *l*, 0.41 ± 0.05 for *m*, −0.02 ± 0.06 for *c*, and 0.903 for the correlation coefficient. The poor correlation and the considerably low *l* value suggest that, as with **5** and **7**, there is a duality of mechanisms, with the operation of the nucleophilic assistance from a solvent molecule and of the intramolecular assistance of the *o*-nitro group. Good correlations are obtained when the 21 solvents are divided into two groups, with each group then analyzed using the extended G–W Equation (3). An analysis of the rate constants of solvolysis in 17 solvents (consisting of the four EtOH–H_2_O mixtures, EtOH, the four MeOH–H_2_O mixtures, MeOH, the three acetone–H_2_O mixtures, and the four TFE–EtOH mixtures) leads to values of 1.10 ± 0.07 for *l*, 0.51 ± 0.02 for *m*, 0.08 ± 0.03 for *c*, and 0.993 for the correlation coefficient (Figure 2) which are very similar to those observed above for solvolyses of **3**, consistent with appreciable nucleophilic participation by the solvent. For solvolyses in solvents with aqueous fluoroalcohol content, values were obtained of 0.07 ± 0.05 for *l*, 1.23 ± 0.15 for *m*, −3.58 ± 0.51 for *c*, and 0.982 for the correlation coefficient. A large negative *c*-value of the *o*-isomer (**1**) in aqueous fluoroalcohols is observed because the experimental *k*_o_ value is the one applying to the other reaction channel. Therefore, the solvolyses of **1** in all of the solvents except aqueous fluoroalcohol, where bond making (*l* = 1.10) is more progressed than bond breaking (*m* = 0.51), are indicated to proceed by a bimolecular pathway (*l/m* = 2.16), reflecting the nucleophilic assistance from a solvent molecule and the intramolecular assistance of the *o*-nitro group (i.e., the *o*-nitrosobenzaldehyde (0.64%) obtained from solvolysis of **1** in 50% EtOH supports the possibility of an intramolecular participation due to assistance of the *o*-nitro group) [3,16]. A plot of log(*k/k_o_*)*_ortho_* for the *o*-isomer (**1**) and log(*k/k_o_*)*_para_* for the *p*-isomer (**3**) in all of the solvents except aqueous TFE shows an excellent correlation (*R* = 0.993), with a slope of 0.931 and an intercept of 0.005 (Equation (6)) (Figure 3).
log (*k_solv_.*/*k_o_*)*_ortho_*= 0.931 log (*k_solv_.*/*k_o_*)*_para_* + 0.007,(6)
*k_ortho_* = *k_intra_* + *k_solv_*.(7)

From Equations (6) and (7), the rate constants for solvolysis of the nucleophilic attack by solvent (*k_solv_*) and intramolecular attack by the *o*-nitro group (*k_intra_*) in aqueous fluoroalcohol (footnotes to Table 1) were calculated as previously described [3,16]. Analysis of rate constants for intramolecular reaction of **1** in aqueous fluoroalcohol solvents by use of the extended Grunwald–Winstein equation shows a negligible dependence on solvent nucleophilicity (0.07 ± 0.05 for *l*, 1.23 ± 0.15 for *m*). Therefore, the data are best analyzed using the simple (original) equation (Equation (2) without the *l* N_T_ term). From the simple G–W Equation (2), the reaction of **1** in aqueous fluoroalcohol solvents is affected by an ionization pathway (0.60 ± 0.08 for *m* and 0.971 for the correlation coefficient). These values are given in Table 4.

The extended G–W Equation (3) was also applied to the solvolyses of **2** and **4**. The analysis of the rate constants of solvolysis of **4** in all of the solvents, except the 97% HFIP, reported the values of 1.78 ± 0.08 for *l* and 0.54 ± 0.04 for *m* (*l/m* = 3.30), reflecting the characteristics of a bimolecular addition–elimination mechanism [12,13,14]. For solvolysis of **2**, an analysis in terms of the extended G–W Equation (3) to the rate constants in all of the solvents gives a poor correlation. Again, the correlations are moderately improved with values of 1.02 ± 0.26 for *l*, 0.35 ± 0.06 for *m*, 0.18 ± 0.08 for *c*, and 0.891 (Figure 4) for the correlation coefficient in 11 solvents (without the fluoroalcohol solvents), and with values of 0.32 ± 0.07 for *l*, 0.71 ± 0.07 for *m*, −0.92 ± 0.14 for *c*, and 0.966 (Figure 5) for the correlation coefficient in all of the fluoroalcohol solvents except 20T–80E (*N*_T_ = 0.08, *Y*_Cl_ = −1.42), and 40T–60E (*N*_T_ = −0.34, *Y*_Cl_ = −0.48) [8,10,11,22,23], involving appreciable nucleophilic solvation, presumably reflecting nucleophilic solvation at the acyl carbon. These correlations led to *l* and *m* values consistent with one set resulting from a bimolecular addition–elimination mechanism (in 11 solvents) and the other from an ionization mechanism (in 10 fluoroalcohol solvents) (Table 4). The *l/m* values also lead to a division of the 21 solvent systems for which both *N*_T_ and *Y*_Cl_ values are available: Into 11, which are assigned to the bimolecular pathway, proceeding through a tetrahedral intermediate, and 10, which are assigned to the ionization pathway (Table 4). The *l* and *m* values for the solvolysis of **2** observed, although the electron withdrawing nitro group favors a bimolecular character, are essentially identical with those for benzoyl chloride, which give evidence for a bimolecular pathway (*l* = 1.27 ± 0.29, and *m* = 0.46 ± 0.07 in 12 solvents) and for an ionization pathway (*l* = 0.47 ± 0.03, and *m* = 0.79 ± 0.02 in 32 solvents) [12,13,14].

Application of the G–W Equation (3) to solvolyses at the *sp*^2^-hybridized carbon of **2** led to the detection of both nucleophilic attack (bimolecular, Equation (4)) and ionization pathways (Equation (5)). In the basic ionization pathway, there is a formation of a nucleophically solvated acylium ion and a chloride ion. This pathway leads to a relatively low *l* value and a relatively high *m* value, such that the *l/**m* ratio is typically appreciably below unity (*l/m* = 0.45). The *l* and *m* values of **2** in Table 4 are very similar to those for the earlier studied halogenoformate esters, which have been shown to solvolyze with the additional step of an addition–elimination pathway being rate determining (Equation (4)) and an ionization pathway (Equation (5)) [24,25,26].

Activation parameters for solvolyses of **1**, **2**, and **3** in several organic hydroxylic solvents are tabulated in Table 2 and 3. The entropies of activation (−21.4 to −30.8 cal∙mol^−1^K^−1^) for the solvolyses of **1** in 80% ethanol, **2** in ethanol, 80% ethanol, and methanol, and **3** in 80% ethanol and 70% TFE are consistent with the bimolecular nature of the proposed rate-determining. In aqueous fluoroalcohol, the activation values for **1** and **2,** except **3,** are slightly less negative entropies of activation (−13.1 to −19.3 cal∙mol^−1^K^−1^). These values are essential similar to those previously observed for the ionization pathway of *o*-carbomethoxybenzyl bromide (**5**, Δ*S^≠^* = −13.3 cal∙mol^−1^∙K^−1^) [3] and *o*-nitrobenzyl *p*-toluenesulfonate (**7**, Δ*S^≠^* = −15.3 cal∙mol^−1^∙K^−1^) in 97% TFE [16].

## 3. Experimental Section

The *o*-nitrobenzyl bromide (**1**) (Sigma-Aldrich, 98%, St. Louis, MO, USA), *o*-nitrobenzoyl chloride (**2**) (Sigma-Aldrich, 97%, St. Louis, MO, USA), and *p*-nitrobenzyl bromide (**3**) (Sigma-Aldrich, 99%, St. Louis, MO, USA) were used as received. All of the substrates were recrystallized from ligroin before using. Solvents were purified as described previously [24]. The rates (**1** and **3**) for the slow solvolytic reactions were followed by potentiometric titrations, and the faster rates (**2**) made using the conductometric method. The kinetic experiments and the products of **1**, **2**, and **3** were carried out as described previously [27,28,29]. The rate constants were obtained by averaging all of the values from, at least, duplicate runs. The products from the reactions of **1** under solvolytic conditions were analyzed after 10 half-lives by gas chromatography (GC-9A, Shimadzu, Kyoto, Japan) using a 2.1 m glass column containing 10% Carbowax 20M on Chromosorb WAW 80/100 as previously described [30]. The solvolysis products of **2** were also determined by GC/mass (Clarus 500 GC-Mass System, Perkin-Elmer, Ion mode EI, Waltham, MA, USA) analysis as described previously [3]. The HPLC system was a Hewlett-Packard 1050 Series instrument (Palo Alto, CA, USA) with a 150 mm × 4.6 mm i.d. Spherisorb ODS reversed-phase column.

## 4. Conclusions

The solvolyses of **1** and **2** in pure and binary solvents except fluoroalcohol, and **3** in all of the solvents, give a satisfactory extended Grunwald–Winstein Equation (3). The sensitivities (*l/m* = 2.16) to changes in *N*_T_ and *Y*_x_ of solvolysis of **1** in all of the solvents, except fluoroalcohol, are very similar to those for **6** [3] and **8** [20,21], which are shown to proceed by a bimolecular character, reflecting nucleophilic assistance from a solvent molecule paralleling the mechanism for **3** (*l/m* = 2.16). However, the solvolyses reaction of **1** in aqueous fluoroalcohol is affected by an ionization pathway (*l/m* = 0.06). And it is shown that the solvolysis of the *o*-isomer (**1**) in all the solvents can be affected not only by nucleophilic assistance paralleling the mechanism of the *p*-isomer (**3**), but also by intramolecular assistance of the *o*-nitro group (Equation (1)).

From the product studies of solvolysis of **2**, the evidence of *o*-substituent participation (*o*-nitro group, **2**) were not found; the ring-closed intermediate formed by intramolecular attack. Accordingly, the solvolyses of **2** without the intramolecular participation by an *o*-nitro group lead to the bimolecular pathway with nucleophilic attack by a solvent at the acyl carbon (*l/m* = 2.92, Δ*S^≠^* = −22.6 ~ −30.8 cal∙mol^−1^∙K^−1^ in EtOH, 80% EtOH, and MeOH) (Equation (4)) and the ionization pathway (*l/m* = 0.45, Δ*S^≠^* = −13.1 ~ −14.3 cal∙mol^−1^∙K^−1^ in 70% TFE, and 70%, HFIP) (Equation (5)).

## Figures and Tables

**Figure 1 ijms-20-04026-f001:**
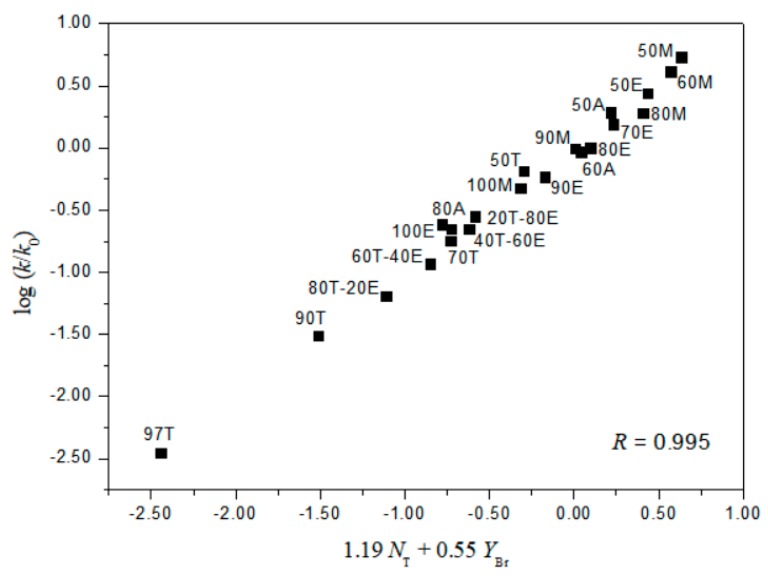
The plot of log (*k/k*_o_)*_p_* vs. 1.19*N*_T_ + 0.55*Y*_Br_ for the solvolysis of *p*-nitrobenzyl bromide (**3**) in various organic solvents at 45.0 °C.

**Figure 2 ijms-20-04026-f002:**
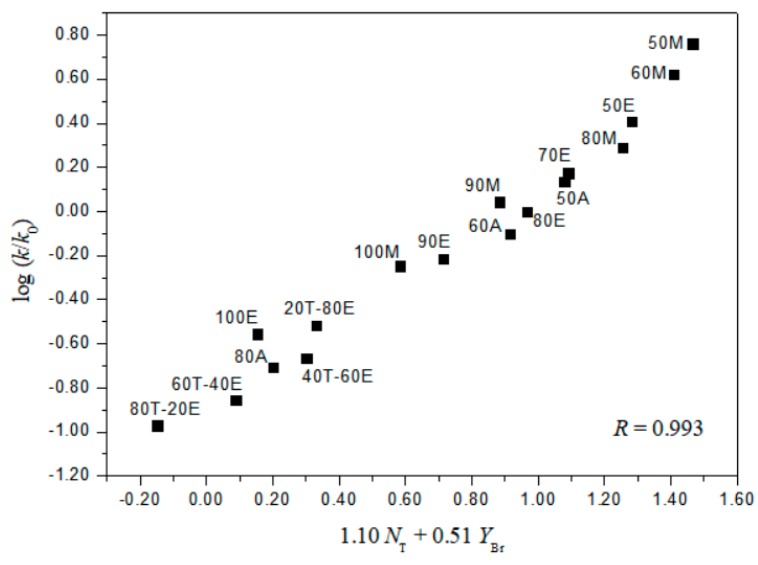
The plot of log (*k/k*_o_)*_o_* vs. 1.10*N*_T_ + 0.51*Y*_Br_ for the solvolysis of *o*-nitrobenzyl bromide (**1**) in various organic solvents at 45.0 °C.

**Figure 3 ijms-20-04026-f003:**
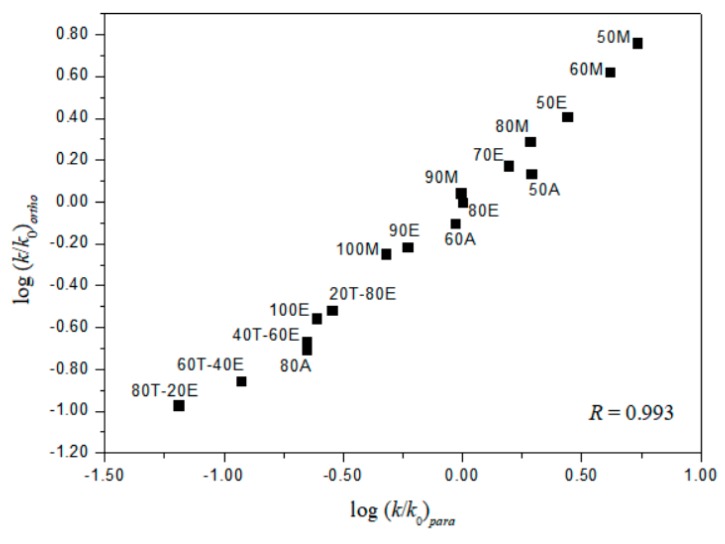
The plot of log(*k*/*k*_o_)*_ortho_* vs. log(*k*/*k*_o_)*_para_* for the solvolyses of **1** and **3** in various organic solvents at 45 °C.

**Figure 4 ijms-20-04026-f004:**
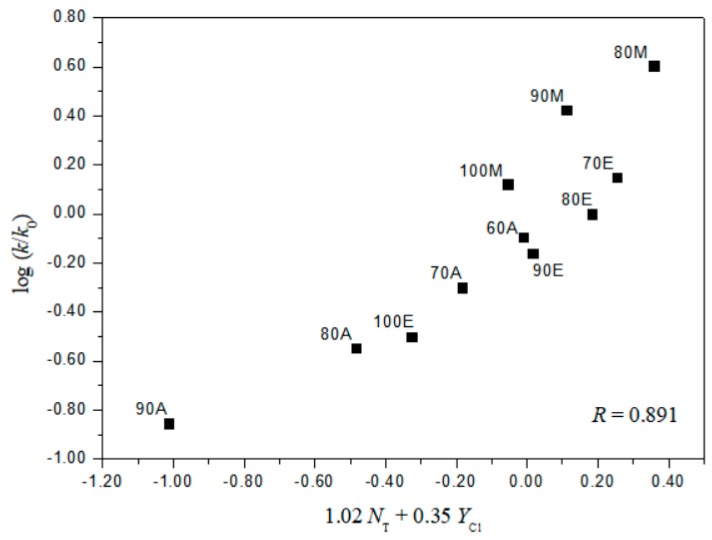
The plot of log (*k/k*_o_)*_o_* vs. 1.02*N*_T_ + 0.35*Y*_Cl_ for the solvolysis of *o*-nitrobenzoyl chloride (**2**) in various organic solvents except all fluoroalcohol solvents at 25.0 °C.

**Figure 5 ijms-20-04026-f005:**
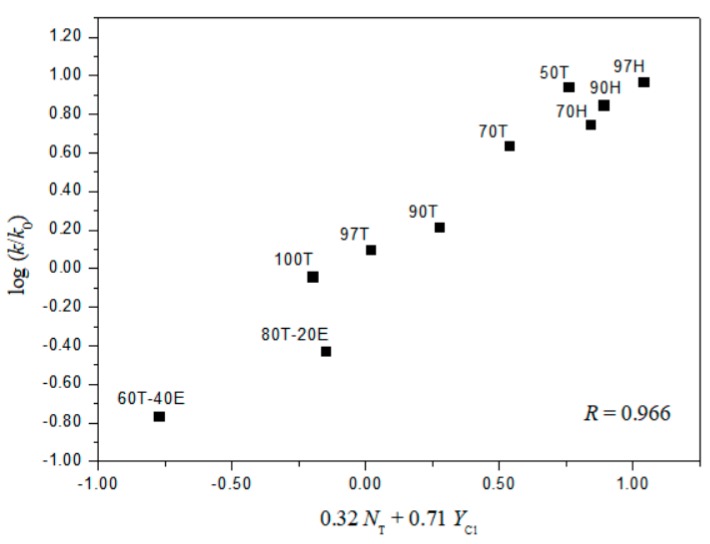
The plot of log (*k/k*_o_)*_o_* vs. 0.32*N*_T_ + 0.71*Y*_Cl_ for the solvolysis of *o*-nitrobenzoyl chloride (**2**) in various fluoroalcohol solvents at 25.0 °C.

**Table 1 ijms-20-04026-t001:** Rate constants (*k*) of solvolysis of *o*-nitrobenzyl bromide (**1**)*^a^* and *p*-nitrobenzyl bromide (**3**)*^b^* at 45.0 °C, and *o*-nitrobenzoyl chloride (**2**)***^c^*** at 25.0 °C, in aqueous binary mixtures and comparisons, and *k*_(**1)**_/*k*_(**3)**_ and *k*_(**2)**_/*k*_(**4**)_ ratios, with corresponding values for the solvolyses of *p*-isomers (**3** and *p*-nitrobenzoyl chloride (**4**)).

Solvent *^d^*	*k*/10^−^^5^ s^−1^ (1) *^a^*	*k*/10^−^^5^ s^−1^ (3) *^b^*	*k*/10^−^^3^ s^−1^ (2) *^c^*	*k*_(1)_/*k*_(3)_*^e^*	*k*_(2)_/*k*_(4)_*^f^*
100% EtOH *^g^*	0.934 ± 0.012	0.834 ± 0.011	1.29	1.1	0.096
90% EtOH	2.06 ± 0.07	2.01 ± 0.08	2.84	1.0	0.079
80% EtOH	3.36 ± 0.06	3.43 ± 0.08	4.09	0.98	0.087
70% EtOH	5.01 ± 0.13	5.33 ± 0.42	5.80	0.94	0.10
50% EtOH	8.58 ± 0.20	9.43 ± 0.13	-	0.91	
100% MeOH *^h^*	1.90 ± 0.03	1.63 ± 0.08	5.41	1.2	0.13
90% MeOH	3.71 ± 0.11	3.37 ± 0.11	10.9	1.1	0.12
80% MeOH	6.54 ± 0.18	6.58 ± 0.17	16.5	0.99	0.13
60% MeOH	14.1 ± 0.2	14.2 ± 0.4	-	0.99	
50% MeOH	19.4 ± 0.2	18.5 ± 0.5	-	1.0	
90% Acetone			0.572		0.099
80% Acetone	0.659 ± 0.011	0.761 ± 0.021	1.16	0.87	0.082
70% Acetone	-	-	2.05		0.098
60% Acetone	2.67 ± 0.09	3.18 ± 0.14	3.30	0.84	0.10
50% Acetone	4.58 ± 0.06	5.64 ± 0.13	-	0.81	
100% TFE	-	-	3.73		5870
97% TFE	0.645 ± 0.052 *^j,k^*	0.0121 ± 0.0013	5.14	53	519
90% TFE	0.939 ± 0.041 *^k^*	0.107 ± 0.030	6.73	8.8	114
70% TFE	1.64 ± 0.03 *^k^*	0.612 ± 0.046	17.8	2.7	29
50% TFE	3.66 ± 0.07 *^k^*	2.25 ± 0.12	36.0	1.6	14
80T–20E *^i^*	0.360 ± 0.022	0.220 ± 0.012	1.53	1.6	8.68
60T–40E *^i^*	0.470 ± 0.072	0.403 ± 0.075	0.702	1.2	0.74
40T–60E *^i^*	0.727 ± 0.031	0.762 ± 0.034	0.667	0.95	0.20
20T–80E *^i^*	1.02 ± 0.02	0.970 ± 0.023	0.914	1.1	0.10
97% HFIP	-	-	38.3	-	21,600
90% HFIP	-	-	28.9	-	
70% HFIP	-	-	22.9	-	

*^a^* Substrate concentration of ca. 5.09×10^−4^ mol·dm^−3^, and titrimetric method. *^b^* Substrate concentration of ca. 4.63×10^−4^ mol·dm^−3^, and titrimetric method.*^c^* Substrate concentration of ca. 5.00×10^−3^ mol·dm^−3^, and conductometric method. *^d^* Volume/volume basis at 25.0 °C, except for TFE–H_2_O and HFIP–H_2_O mixtures, which are on a weight/weight basis. *^d^* Values from Reference 50. *^e^* The ratios of solvolysis rate constant for the *o*-isomer (**1**) and *p*-isomer (**3**). *^f^* The ratios of solvolysis rate constant for the *o*-isomer (**2**) and *p*-isomer (**4**), and Reference [8]. *^g^* A.*^i^* T–E is TFE–ethanol mixtures. *^j^* Extrapolated value, obtained by applying the Arrhenius equation to rate constants measured at higher temperatures (Table 2). *^k^* Calculated from *k*_intra_ = *k*_ortho_ − *k*_solv_ (Eqaution 7). The rate constants for solvolysis of the intramolecular attack by the *ortho* nitro group (*k*_intra_) in aqueous TFE (At 97% TFE, 90% TFE, 70% TFE, and 50% TFE, values of 0.6267×10^−5^ sec^−1^, 0.804×10^−5^ sec^−1^, 0.954×10^−5^ sec^−1^, and 1.35×10^−5^ sec^−1^, respectively.

**Table 2 ijms-20-04026-t002:** Rate constants (*k*) and activation parameters (Δ*H^≠^* and Δ*S*^≠^) for solvolysis of **1** and **3** in aqueous binary mixtures at various temperatures.

*k*/10^−6^ s^−1^					
Temp. (°C)	*o*-isomer (1)			*p*-isomer (3)	
	80% EtOH *^a^*	70%TFE *^b^*	97% TFE *^b^*	80% EtOH *^a^*	70% TFE *^b^*
45.0	3.36 ± 0.05	1.64 ± 0.003	0.645*^e^*	3.43 ± 0.08	0.612 ± 0.046
58.1	12.0 ± 1.0*^c^*	-	2.08 ± 0.007	12.1 ± 0.5	-
62.5	-	9.08 ± 0.03 *^d^*	5.43 ± 0.005	18.8 ± 1.0	-
68.0	29.8 ± 1.1	16.9 ± 0.08	6.95 ± 0.008 *^f^*	30.7 ± 1.0	6.63 ± 0.28
73.0	-	27.5 ± 0.07	9.97 ± 0.004	-	8.83 ± 0.22
83.0	-	-	-	-	21.3 ± 1.5
Δ*H^≠^*_318.15_ (kcal/mol)	19.8 ± 0.1	21.3 ± 0.6	21.5 ± 0.5	19.9 ± 0.2	18.5 ± 0.5
−Δ*S*^≠^_318.15_ (cal/mol·K)	21.4 ± 0.2	18.1 ± 2.0	19.3 ± 1.6	21.0 ± 0.7	28.9 ± 1.7

*^a^* 80% EtOH prepared on a volume/volume basis, at 25.0 °C and 70% TFE and 97% TFE prepared on a weight/weight basis. *^c^* At 58.0 °C. *^d^* At 62.0 °C. *^e^* Extrapolation value. *^f^* At 68.1 °C.

**Table 3 ijms-20-04026-t003:** Rate constants (*k*) and activation parameters (Δ*H^≠^* and Δ*S*^≠^) for the solvolysis of **2** in aqueous binary mixtures at various temperatures.

Solvent *^a^* (%)	Temp. (°C)	*k*/10^−3^*^b^*(s^−1^)	Δ*H^≠^*_298.15_*^c^*(kcal/mol)	−Δ*S*^≠^_298.15_*^c^*(cal/mol·K)
100EtOH	25.0 20.0 15.0 10.0	1.29 0.833 0.571 0.410	12.2 ± 0.7	30.8 ± 2.4
80EtOH	25.0 20.0 15.0 10.0	4.09 2.61 1.72 1.11	14.0 ± 0.2	22.6 ± 0.8
100MeOH	25.0 20.0 15.0 10.0	5.41 3.74 2.58 1.76	11.9 ± 0.08	28.9 ± 0.3
70TFE	25.0 20.0 15.0 10.0	17.8 10.4 6.91 3.95	15.9 ± 0.6	13.1 ± 2.1
70HFIP	25.0 20.0 15.0 10.0	22.9 14.4 8.99 5.47	15.4 ± 0.06	14.3 ± 0.2

*^a^* 80% EtOH prepared on a volume/volume basis, at 25.0 °C and 70% TFE and 70% HFIP prepared on a weight/weight basis.*^b^* Values are average of two or more runs. *^c^* With associated standard error.

**Table 4 ijms-20-04026-t004:** Correlation of the rate constants for the solvolyses of **1**, **2**, **3**, and **4**, using the extended G–W Equation (3).

Substrate	*n ^a^*	*l ^b^*	*m ^b^*	*c ^c^*	*r ^d^*	*l/m ^e^*
1	21 *^g^*	0.66 ± 0.07	0.41 ± 0.05	0.02 ± 0.06	0.903	-
	17 *^h^*	1.10 ± 0.07	0.51 ± 0.02	0.08 ± 0.03	0.993	2.16
	04 *^i^*	0.07 ± 0.05 ~0	1.23 ± 0.15 0.60 ± 0.08*^l^*	−3.58 ± 0.51 −2.22 ± 0.23	0.997 0.971	0.06 ~0
2	23 *^g^*	0.23 ± 0.11	0.35 ± 0.08	−0.03 ± 0.11	0.779	-
	11 *^j^*	1.02 ± 0.26	0.35 ± 0.06	0.18 ± 0.08	0.891	2.94
	10 *^k^*	0.32 ± 0.07	0.71 ± 0.07	0.92 ± 0.14	0.966	0.45
3	21 *^g^*	1.19 ± 0.03	0.55 ± 0.02	0.09 ± 0.02	0.995	2.16
4*^f^*	34 *^g^*	1.78 ± 0.08	0.54 ± 0.04	0.11 ± 0.37	0.969	3.30

*^a^ n* is the number of solvents. *^b^* With associated standard error. *^c^* Accompanied by standard error of the estimate. *^d^* Correlation coefficient. *^e^ l/m* values of the extended G–W Equation (3). *^f^* Results obtained using rate constant data from References [4,12,13,14,15]. *^g^* All available solvents. *^h^* All the solvents excluding the data points in the TFE (aq) solutions. *^i^* Just the TFE (aq) data points. *^j^* All the solvents excluding the data points in all the TFE (aq) and HFIP (aq) solutions. *^k^* The data points in all the TFE (aq) and HFIP (aq), excluding 20T–80E and 40T–60E solutions. *^l^* Value obtained using rate constants of the simple G–W Equation (2).

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
