# Peer review of "The Effect of the ortho Nitro Group in the Solvolysis of Benzyl and Benzoyl Halides"

_ijms, 2019, doi:10.3390/ijms20164026_

Round 1
Reviewer 1 Report
In the manuscript entitled "The Effect of ortho Nitro Group in the Solvolysis of Benzyl and Benzoyl Halides" the authors report kinetic data for the solvolysis of o-nitrobenzyl bromide and o-nitrobenzoyl chloride using different solvent conditions under various temperatures.
On the basis of the overall data the authors suggest that the solvolysis o-nitrobenzyl bromide but not that of o-nitrobenzoyl chloride can be affected by intramolecular assistance of the o-nitro group. The mechanistic deductions on solvolysis of the analyzed compounds are based on a model that takes into account the nucleophilicity and the ionizing power of the used solvents.
The work adds some new mechanistic information about the reactivity of the studied molecules and could be accepted after the following revisions:
1. A picture containing the chemical structures of the molecules cited in the text should be added.
2. The method used to obtain the data reported in the tables 2 and 3 should be described in the experimental section.
3. The values of the 4th column (k/ 10-3 s-1 (2)) of the table 1 and that of the 7th and 8th rows of the table 2 are reported without a statistical evaluation of the errors.
4. For clarity, the points reported in the figures 1, 2, 3, 4 and 5 should be marked by an acronym indicating the solvent used.
5. The quality of the figures 1-7should be improved.
6. A more effective conclusion about the data obtained should be added. Can the reported results for 1 and 2 be useful in some way to make predictions for other species?
7. In the experimental section the manufacturer and the degree of purity of the molecules used in the experiments should be reported.
8. Some pieces of the manuscript should be revised (rows 27-29: “… the o-isomer … solvents is known to undergo solvolytic reactions many times more rapidly than its the p-carbomethoxybenzyl bromide. Row 70 “In an attempt to define better the reaction pathways for reactions of 1 and 2 under solvolytic..” Rows 202-204 “An analysis of the rate constants of solvolysis of 4 in all of the solvents except only 97% HFIP has been previously reported values of 1.78±0.08 for l and 0.54±0.04 for m” Rows 266-267 “From the product studies of solvolysis of 2, evidence of o-substituent participation..., are not found the ring-closed products of the intermediate formed by intramolecular attack” etc)
Author Response
To Reviewer1
Thank you so much for reviewing our article in this time.
I answered to your correction number each by each.
The picture containing the chemical structures is benzoyl chloride. This molecule is basic compound of the research one in this article. So we would like to show the scheme of the possible mechanism for 3 compound by using the basic compound. Finally we added the small title for scheme. please check again. The obtaining method for rate constant were already showed in the experimental section. We described by slow and fast reactions separately. So we added our molecule symbols to make clearly. We added statistical evaluation of the errors in the 7th and 8th row of the table 2. However, the 4th column of the table 1, these values of 2 were obtained by the conductivity method which is measured at regular time intervals, and the rate is directly calculated by using a curve equation with the conductivity values in about 100 sections. The rates is calculated by averaging the two experiments, so we did not calculate the standard deviations because of no meaning. While the rates of 1 and 3 were obtained by using titration method that the rates at 5 ~ 7 points were obtained at different time intervals, and were averaged these values. So we calculated and indicated the standard deviations for 1 and 3 compounds. Hopefully our explanation is enough to understand the difference between methods to obtain the rates. We indicated all plot with the corresponding solvent names. We tried again to improve the quality of all figures by using the different way. Actually, our analysis methods(G-W equation and activation parameters) are the considering of numbers which is obtained by the calculation statistically. So without the results, we could not predict in detail. I have never seen a paper in which researchers in this area predicted mechanisms for other compounds without any results. This is not easy to establish the standard values in even series compounds. I hope you understand this research area. We added the company name and purity of compounds studied in this article. We tried to revised well again. please check again.
Thank you so much again for reviewing our manuscript.
Sincerely,
Kyoung-Ho Park
Reviewer 2 Report
The manuscript describes a kinetic study of the solvolyses of two nitrobenzyl bromide isomers and of o-nitrobenzyl chloride in a diverse set of organic hydroxylic solvents, some of them pure and the large majority in binary aqueous and ethanolic mixtures of varying percentage.
The aim of the work is correctly addressed in the chosen title, the length and depth of the abstract are correct, and keywords are appropriate.
The experimental work is of good quality and the G-W methodology used has long been widely accepted.
Results are interesting and conclusions are correctly drawn.
The length of the manuscript, sections, number of figures, equations and references is adequate albeit extensive self-referencing (13 out of 30, that is 43%!!). This is hard to understand even if these research topics are indeed limited to a small number of researchers. In other journals, authors would have to justify referencing each of their own papers. Anyway, I do not think the majority of self-references are needed to support the work proposed for publication.
English sentencing is average in most of the manuscript, but several sentences need to be corrected/re-written, e.g.,
In the abstract, line 11: “reactions of 1 were solvolyzed similar (??) to those observed for (??) 3, and the reactions of 2 about ten times more slowly than those (??)”
In the abstract, line 13: “the reactivity for 2 was kinetically more reactive than 4” – reactivity “of” instead of “for” and repetition – “reactivity (…) more reactive“ …
In the abstract, line 16:" The role of the ortho-nitro group as an intramolecular nucleophilic assistance (internal nucleophile)" - replace "assistance" by "assistant"?
Main text, line 29:"Relatively high reactivity of the o-isomer" - replace "high" by "higher"
Main text, line 36: “In the consideration of the solvolyses of benzoyl chloride and p-nitrobenzoyl chloride (4) [4], they have been extensively studied” - what is the meaning of this sentence?
Main text line 57: “the mechanisms of these reactions have been studied less kinetically“ - this does not make any sense...
Main text line 70 “In an attempt to define better the reaction pathways” - ???
N.B.: The above sentences include only a part of the sentences that needed to be corrected.
In table 2 the activation parameters must be given along with their uncertainties
This reviewer considers that one should follow IUPAC and write: parameter/units, e.g. 106k /s-1. Only then will the numbers be mathematically correct. The same is true for X/cal mol-1 K-1 instead of X (cal/mol.K). The final format is, of course, left to the authors and editor.
The size of figures 1 to 3 should be the same. The number of significant digits on the axis’ tick marks should be 3 (or 2 decimals) according to log k uncertainties and the values indicated for parameters l and m in the x-axis, e.g., 1.10 and 0.51.
I do not agree with the use of “r” as a measure of a fit’s quality. Only “R2” or, better still “R2adj” is statistically significant. Anyway, it is mandatory to use the letter “R” and not “r” when showing multiple regression results.
Author Response
To Reviewer2
Thank you so much for reviewing our article in this time.
We corrected all you indicated in review material.
I appreciate your kind review again.
Sincerely,
Kyoung-Ho Park